# Synthesis, Structural Determination, and Antioxidant Activities of Acyclic and Substituted Heterocyclic Phosphonates Linearly Linked 4-hydroxy-2(1*H*)-quinolinone

**DOI:** 10.3390/molecules27185960

**Published:** 2022-09-13

**Authors:** Mohamed M. Hassan, Mona H. Alhalafi

**Affiliations:** 1Chemistry Department, Faculty of Education, Ain Shams University, Roxy, Cairo 11711, Egypt; 2Department of Chemistry, College of Science Al-Zulfi, Majmaah University, P.O. Box 66, Al-Majmaah 11952, Saudi Arabia; m.alhalafi@mu.edu.sa

**Keywords:** enaminone, phosphorus sulfides, Lawesson’s reagent, organophosphorus compounds, antioxidant

## Abstract

The chemical reactivity of 3-[(*E*)-3-(dimethylamino)-2-propenoyl]-4-hydroxy-1-methy-2(1*H*)-quinolinone (**1**) towards some phosphorus reagents was studied. The enaminone **1** was cyclized into pyranoquinolinylphosphonate **2** via treatment with diethyl phosphite in basic medium. However, its reaction with triethoxy phosphonoacetate gave the substituted oxopyranylphosphonate **3**. Using the same reaction conditions, both thioxopyridinylphosphonate **4** and oxopyranylphosphonate **5** were produced via a reaction of enaminone **1** with both diethyl 2-amino-2-thioxoethylphosphonate and diethyl vinylphosphonate, respectively, in low yields. In addition, the two novel oxopyridinylphosphonates **6** and **7** were obtained by treatment of enaminone **1** with a diethyl cyanomethylphosphonate reagent. Two oaxathiaphosphininyl derivatives, **8** and **9,** were obtained by treatment of the enaminone **1** with *O, O*-diethyl dithiophosphoric acid under different reaction conditions. Diazaphosphininyl **11** and oxazaphosphininyl **12** derivatives were obtained in excellent yields using a *P*-phenylphosphonic diamide reagent under different reaction conditions. The treatment of the enaminone **1** with phosphorus pentasulfide produced the non-phosphorylated product thioxothiopyranoquinolinone **13**. Finally, the enaminone was turned into oxathiaphosphininyl **14** using Lawesson’s reagent. The possible reaction mechanisms of the formation of these products were discussed. The structures of newly isolated products were established by elemental analysis and spectral tools. The compounds were evaluated for their antioxidant activities.

## 1. Introduction

Briefly, 2-Quinolinone compounds are a class of bioactive heterocyclic systems that have received a lot of attention due to their considerable pharmacological actions (antioxidant, anti-inflammatory, anti-malarial, antibacterial, and anticancer effects) [1,2,3,4,5,6,7]. The discovery of significant structural diversity in phosphorus heterocycles sparked rapid development in the chemistry of these heterocycles during the past few decades. These substances have uses in the medical and agricultural industries, including anti-inflammatory, analgesic, and anti-cancer action [8,9]. We anticipated that combining phosphorus heterocycles with a 2-quinolinone moiety in a single molecular frame may boost biological features due to the synthetic and biological benefits of these two compounds. To the best of our knowledge, no literature reports the synthesis of phosphorus heterocycles obtained from 4-hydroxy-2-quinolinone. In recent studies conducted by us and others, bioactive phosphorus compounds containing heterocycles were produced [10,11]. Enaminones, on the other hand, are chemical substances composed of an amino group connected to a carbonyl group via a C=C bond. The ambident nucleophilicity of enamines and the ambident electrophilicity of enones combine to form these adaptable synthetic intermediates. Enaminones react with a variety of electrophilic and nucleophilic reagents to generate a number of varied heterocycles thanks to their diverse chemical reactivities [12,13,14]. The current study aims to synthesize novel phosphorus heterocycles using a 2-quinolinone system and explores these compounds’ potential antioxidant effects.

## 2. Results and Discussion

### 2.1. Synthetic Strategies

In this study, 3-[(*E*)-3-(dimethylamino)-2-propenoyl]-4-hydroxy-1-methy-2(1*H*)-quinolinone (**1**) [15] was treated with some phosphorus reagents before being utilized as a starting material to create some unique phosphorus heterocycles connected to a 2-quinolinone ring. The new diethyl 5,6-dihydro-6-methyl-4,5-dioxo-4*H*-pyrano[3,2-*c*]quinolin-2-yl-2-phosphonate (**2**) was produced in a low yield of 35% when 3-[(*E*)-3-(dimethylamino)-2-propenoyl]-4-hydroxy-1-methy-2(1*H*)-quinolinone (**1**) was allowed to react with diethyl phosphite in toluene containing a few drops of 1,8-Diazabicyclo[5.4.0]undec-7-ene (DBU) as a basic catalyst (Figure 1). Strong absorption bands were visible in the IR spectrum of **2** at 1650 (C=O _pyrone_), 1615 (C=C), 1228 (P=O), and 1026 (P-O-C) cm^−1^. Product **2**’s ^31^P-NMR spectra had a singlet at δ 21.1 ppm [16]. A triplet at δ 1.20 (6H, *J* = 7.3 Hz) and a quartet at δ 4.15 (4 H, *J* = 7.3 Hz) in its ^1^H-NMR spectrum were attributed to the diethoxyphosphoryl group connected to the quinolinone ring, respectively. At δ 6.20 ppm, the H-3 proton connected to the pyran ring also showed up as a singlet. Additionally, its mass spectrum showed the anticipated molecular ion peak at *m*/*z* 363 (M^+^, 10%). A rational justification for the synthesis of compound **2** is presented in (Figure 1). The phospha-Michael nucleophilic addition process results in the elimination of the dimethylamine moiety to yield the non-isolable intermediate **A,** which underwent an air-free, spontaneous intramolecular cycloaddition to form the intermediate **B**. Sequentially, the later intermediate underwent hydrogen elimination, affording the expected quinolinonyl phosphonate **2**.

Enaminone **1** was used to accomplish reactions with phosphonyl reagents by combining compound **1** with the reagent while it was still dispersed in acetic acid/H_2_SO_4_. The reaction mixture was heated for the appropriate amount of time (~10 h, TLC) at 100 °C to produce the appropriate phosphonates. As a result, diethyl 6-(1,2-dihydro-4-hydroxy-1-methyl-2-oxoquinolin-3-yl)-2-oxo-2H-pyran-3-yl-3-phosphonate (**3**) was produced in a moderate yield 50% when enaminone **1** and triethoxy phosphonoacetate were combined. In order to create **3**, the carbanion-C must have first attacked the allylic system while also losing the HNMe_2_ moiety. The phosphonate **3** was produced by the further extrusion of the appropriate alcohol moiety from the resultant intermediate **C** (Figure 2). The structure of the isolated product was determined using spectral data. The absorption bands for compound **3** were at 1670 (for C=O_pyrone_), 1250 (for P=O), and 1032 (for P-O-C). Additionally, compound **3**’s ^31^P-NMR spectra revealed a singlet at δ 24.2 ppm that was caused by a phosphonate group. The diethoxy moiety of this compound was visible in its ^1^H-NMR at δ 1.05 and 1.35 (t’s, 6H, 2CH_3_) and 3.75-3.80, 4.13-4.15 (m, 4H, CH_2_) ppm, whereas the H-5 proton of the pyrone ring was visible as a doublet at δ 6.50 ppm, with a coupling constant of 5.6 Hz. Furthermore, the doublet (*J*_PC_ = 174 Hz) produced by the C-P was visible in the ^13^C-NMR spectrum of **3** at δ 118.4 ppm, whereas the C=O_pyrone_ was present at δ 167.6 ppm. Compound **3**’s mass spectrum revealed a molecular ion peak at *m*/*z* 405 (M^+^, 20%), confirming its proposed structure.

Similarly, enaminone **1** was allowed to react with diethyl 2-amino-2-thioxoethylphosphonate, and a 55% yield of the diethyl[6-(4-hydroxy-1-methyl-2-oxo-1,2-dihydroquinolin-3-yl)-2-thioxo-1,2-dihydropyridin-3-yl]phosphonate (**4**) was produced (Figure 2). The spectroscopic characteristics of Structure 4 (δ*_P_* = 26.4 ppm) allowed for its identification. A faint band at 1225 cm^−1^ in the IR spectra of substance **4** indicated the existence of a thio-carbonyl group. The IR spectrum also showed strong bands at 1230 (P=O) and 3330 (NH). The two ethoxy [P(OEt)_2_] protons were visible at δ 1.18 ppm in the ^1^H- NMR spectrum of compound **4**, while the imino proton (HN) was visible at δ10.50 ppm. The distinct ethoxy group was identified by ^13^C-NMR spectroscopy. Additionally, the chemical shift at δ 26.4 ppm of the proposed structures’ ^31^P-NMR spectra matched the chemical shift.

Diethyl vinylphosphonate treatment with **1** produced the beneficial diethyl[6-(4-hydroxy-1-methyl-2-oxo-1,2-dihydroquinolin-3-yl)-2*H*-pyran-3-yl]phosphonate (**5**) with a yield of 40%. One-step cycloaddition was thought to generate compound **5** while simultaneously eliminating the HNMe_2_ molecule (Figure 2). The protons of the methylene group (H_2_C/pyran)) were exhibited as two doublets of doublets at δ 5.00 and δ 5.15 ppm in the ^1^H-NMR spectra, indicating that they were not magnetically equivalent. They formed a coupling with the phosphorus atom (*J*_PH_ = 8.8 Hz) after coupling with each other (*J*_HH_ = 7.6 Hz). An additional signal in the ^13^C-NMR spectra of compound **5** at δ 61.8 (d, *J*_PC_ = 10.8 Hz) confirmed the presence of CH_2_/pyran in that substance. Due to the chemical shift of C-P, a doublet (*J*_PC_ = 174 Hz) was also seen for (C−3′) at δ 123.2 ppm. The phosphorus atom signal at δ 25.5 ppm in the ^31^P-NMR spectra of this molecule supports the proposed structure. The mass spectra and chemical formula were very identical.

The substituted diethyl [6-(4-hydroxy-1-methyl-2-oxo-1,2-dihydroquinolin-3-yl)-2-oxo-1,2-dihydropyridin-3-yl]phosphonate (**6**) was obtained in a 48% yield by reacting compound **1** with diethyl cyanomethylphosphonate under the same strong acidic circumstances (Figure 3). The presence of an acidic medium was responsible for the reduction in the intermediate’s **E** cyano-group. Strong absorption bands at 2980 (C-H_aliphatic_), 1720 (C=O_amide_), and 1076 (P-O-C) cm^−1^ were visible in compound 6’s IR spectra. The ethoxy group’s distinctive triplet (δ = 1.13 ppm) and quartet (δ = 3.82 ppm) were present in the ^1^H-NMR spectra of compound **6**. Additionally, it revealed a brand-new singlet for the NH proton at δ 10.2 ppm. The ^13^C-NMR spectra of **6**, which showed the specific carbon atoms at d 15.0 (CH_3_), 64.0 (CH_2_), and 164.5 (C=O_amide_), were also used to support the structure of **6**. Additionally, a chemical shift in its ^31^P-NMR signal was seen at δ 24.0 ppm. In (Figure 3), a conceivable method for the production of **6** is put forth.

Additionally, under thermal conditions, the interesting novel compound diethyl 4-(dimethylamino)-1,2-dihydro-6-(1,2-dihydro-4-hydroxy-1-methyl-2-oxoquinolin-3-yl)-2-oxopyridin-3-yl-3-phosphonate (**7**) was smoothly produced in 50% yield by reacting enaminone **1** with diethyl cyanomethylphosphonate (formed in situ) (Figure 3). The use of spectrum tools and elemental analyses supported the structure of compound **7**. The OH proton was seen in the ^1^ H-NMR spectra of compound **7** at δ 11.10 ppm, as well as three other forms of methyl protons at δ 1.30 (3H, t, EtO), 2.56, 2.84 (6H, s, 2NMe_2_), and 3.58 (3H, s, N-Me_quinolinone_). Additionally, the pyridinone ring’s (H4′) proton vanished, leaving only the (H5′) in the well-known aromatic region. Briefly, 2NMe_2_ (δ 43.4 and 46.1) and 2EtO (δ 13.5 and 60.1) ppm, as more distinctive carbon atoms, were also seen in the ^13^CNMR spectra. Compound **7**’s ^31^P-NMR chemical shift showed a singlet at 25.0 ppm, which agreed with its chemical shift value. Additionally, the molecular ion peaks at *m*/*z* 447 in compound **7**’s mass spectra confirmed its suggested structure. Evidently, the initial nucleophile attacks the phosphorus reagent on the allylic system’s *β*-carbon atom to produce the intermediate **F** without eliminating the dimethylamine molecule, which would be rare [17]. The 1,3-proton shift in the latter intermediate was followed by cyclization when enolic OH attacked the phosphonate moiety giving rise to the non-isolable **H** intermediate which underwent *Dimorth* rearrangement and auto-oxidation to afford the final product **7** (Figure 3).

In a similar way, compound **1** was permitted to interact with *O*,*O*-diethyl dithiophosphoric acid to produce two products that could be separated under the appropriate reaction conditions. The compound 3-[4-(dimethylamino)-2-ethoxy-2-oxido-4H-1,3,2-oxathiaphosphinin-6-yl]-4-hydroxy-1-methylquinolin-2(1*H*)-one (**8**) was the first product, with a yield of 40%, and 3-(2-ethoxy-2-oxido-4H-1,3,2-oxathiaphosphinin-6-yl)-4-hydroxy-1-methylquinolin-2(1*H*)-one (**9**) was the second product in a 45% yield (Figure 4). The isolated products **8** and **9** were in full agreement with their spectroscopic data. Compound **8**’s ^1^H-NMR spectrum showed the three singlet’s that are distinctive of it (9H, NMe^,^S), which also resonate at δ 28.5, 41.1, and 44.1 ppm in its ^13^C-NMR spectra. Additionally, indicating their presence in cis-form were the two protons of the 1,3,2-oxathiaphosphinine ring, which showed up as two doublets at δ 6.10 and 6.80 ppm with a coupling constant of 7.2 Hz.

The ^1^H-NMR spectrum of compound **9** revealed that the two methylene group protons of (H_2_C/oxathiaphosphinine) were non-magnetically equivalent and were exhibited at δ 4.33and 4.35 ppm, in addition to the characteristic one singlet at δ 3.10 ppm for (3H, -NCH_3quinolinone_). An additional indication of (CH_2oxathiaphosphinine_) in **9** was a signal at δ 61.0 in its ^13^C-NMR spectra. The mass spectrometry was an excellent tool to prove the proposed structures for both compounds **8** and **9** and showed the molecular ion peaks at *m*/*z* 396 and 353, respectively.

Evidently, the initial nucleophile attacks the thiol group of the phosphorus reagent on the *β*-carbon atom of the allylic system to generate the intermediate **I** without the unique action of removing the dimethylamine molecule [17]. The 1,3-proton shift in the later intermediate was followed by the removal of the ethanol molecule by the enolic OH attacking the phosphonate moiety forming the first product **8** (Figure 4). According to the reaction mechanism depicted in (Figure 4), compound **9** was appropriately generated by first attacking the *β*-carbon atom of the allylic system of enaminone **1** with the P-reagent, along with the loss of the dimethylamine moiety, to produce the intermediate **K**. This intermediate underwent a 1,4-proton shift, then an intramolecular cyclization further eliminated the ethyl alcohol molecule from the intermediate **L** to afford the expected product **9**.

In an effort to increase the reaction yield, equimolar amounts of compound **1** and *P*-phenylphosphonic diamide were added to dry dioxane containing a few drops of triethylamine as a basic catalyst, then heated under reflux to give a mixture of 4-hydroxy-1-methyl-3-(2-oxido-2-phenyl-1,2-dihydro-1,3,2-diazaphosphinin-4-yl)quinolin-2(1*H*)-one (**11**) and 4-hydroxy-1-methyl-3-(2-oxido-2-phenyl-3,4-dihydro-2*H*-1,3,2-oxazaphosphinin-6-yl)quinolin-2(1*H*)-one (**12**) in excellent yields with no isolation of compound **10** at any formation percentage. There is no evidence for the existence of any additional N-Me,S except N-Me_quinlinone_ at the appropriate spectral region, according to spectrum data, particularly ^1^H-NMR and ^13^C-NMR. Compound **11**’s IR spectrum showed that P=O, C=N, and NH, respectively, have absorption bands at 1250, 1650, and 3230–3175 cm^−1^. While compound **12** had the typical absorption bands for (P=O), (P-O-C), and NH at 1253, 1131, and 3200–3150 cm^−1^, respectively. The ^1^ H-NMR spectrum for both compounds **11** and **12** showed the two singlets ascribable to both NH and –NCH_3quinolinone_ groups. The two analogous protons’ diazaphosphinine ring (H-5′ and H-6′) was seen in compound **11’s** ^1^H-NMR spectra as a doublet of a doublet at δ 6.94 and 7.04 ppm, respectively. In addition to the triplet signal attributed to (H-5′) at about 6.60 ppm, the ^1^H-NMR spectrum of compound **12** revealed that the protons of the methylene group (H_2_C_oxazaphosphinine_) were not magnetically equivalent and appeared as two doublets of doublets at about δ 4.05 and 4.17 ppm. Another indicator of the presence of CH_2oxazaphosphinine_ was a doublet signal at δ 68.6 ppm in the ^13^C-NMR spectra. All of the carbons’ distinctive signals could be seen in the ^13^C-NMR spectra, but those corresponding to the diazaphosphinine **11** and oxazaphosphinin **12** rings stood out in particular. The carbon atoms that were bound to the phosphorus atoms were of special interest since they resonated as doublets with the correct coupling constants between d 134.0 and 136.0 ppm. Singlets were detected in the ^31^P-NMR chemical shift at δ 31.50 and 28.50 ppm, respectively, which matched the values for the 1,3,2-diazaphosphinine and 1,3,2-oxazaphosphinine chemical shifts. Additionally, the mass spectrum, which demonstrated the proper molecular ion peaks at m/e 365 and m/e 368, respectively, corroborated the structure of compounds **11** and **12**.

Compounds **11** and **12** were correctly created by first attacking the allylic system of enaminone 2’s carbon atom with the P-reagent and losing the dimethylamine moiety to produce the intermediate **M** in accordance with the reaction process given in (Figure 5). Compound **11** was produced with a superb yield of 80% through an intramolecular nucleophilic attack of the amino group on the carbonyl group with the elimination of water [18]. In relation to the synthesis of compound **12**, which was separated with an excellent reaction yield of 75%, an ammonia molecule was eliminated from the intermediate **N** via intramolecular cyclization following a 1,4-proton shift for the prior intermediate **M**.

It was reported that certain bifunctional compounds might change into intriguing phosphorus heterocycles when exposed to phosphorus sulfides [19,20]. This inspired us to investigate how enaminone **1** reacts with some phosphorus sulfides.

Orange crystals of the expected non-phosphorylated product 6-methyl-4-thioxo-4*H*-thiopyrano[3,2-*c*]quinolin-5(6*H*)-one (**13**) were produced in dry toluene by the reaction of phosphorus pentasulfide with compound **1** in good yield (70%) yield (Figure 3). According to the experimental section, all spectrum tools (^1^H-NMR, ^13^C-NMR, FT-IR, and MS) were used to confirm the newly synthesized compound **13**. According to the chemical mechanism depicted in (Figure 6), compound **13** was created by the thionating of both the enone carbonyl and hydroxyl groups, then cyclizing intramolecularly and removing the dimethylamine molecule with no thionation effect on the quinolinyl carbonyl group via both **O** and **P** intermediates.

Intriguingly, the reaction of enaminone **1** with Lawesson’s reagent (LR) in dry toluene resulted in the beneficial compound 3-(4-(dimethylamino)-2-(4-methoxyphenyl)-2-sulfanylene-4*H*-1,3,2-oxathiaphosphinin-6-yl)-4-hydroxy-1-methylquinolin-2(1H)-one (**14**) in (Figure 6). According to (Figure 5), compound **14** was thought to be created via a one-step cycloaddition mechanism. The protons of methyl groups were seen in the ^1^H-NMR spectrum of compound **14** at δ 2.90, 3.15 (2NCH_3_), 3.32 (NCH_3quinolinone_), and 3.80 (OCH_3_) ppm. A signal at δ 52.3 ppm in the ^13^C-NMR spectrum of **14** confirmed the presence of a methoxy group, whereas signals for the dimethylamino group occurred at δ 32.1 and 33.8 ppm. The ^31^P-NMR spectra of compound **14** also contained a signal at δ 28.5 ppm. Additionally, compound **14**’s mass spectrum confirmed its structure by displaying the proper molecular ion peak at m/e 474.

### 2.2. Pharmacology

#### Antioxidant Activity

Two in vitro techniques were used to assess the synthetic compounds’ antioxidant capacities in order to compare the results and identify some correlations between structure and antioxidant activity. At different concentrations of 50, 75, and 100 g/mL, the evaluation research was conducted. An efficient method for assessing the radical scavenging capacity of particular compounds or extracts is the DPPH (2,2-diphenyl-1-picrylhydrazyl) radical scavenging activity evaluation, which is a standard test in antioxidant activity research [21,22]. A freshly created DPPH solution has a deep purple hue and peaks at 517 nm for absorption. Usually, when an antioxidant is present in the medium, this purple color fades. As a result, an antioxidant molecule can quench the DPPH free radical (by giving it hydrogen atoms or electrons) and turn it into the colorless 2,2-diphenyl-1-picrylhydrazine, which reduces absorbance. Therefore, the compound’s antioxidant activity is more effective the faster the absorbance decreases. We compared and assessed the percentage activity of ethanolic solutions of the produced drugs (Table 1). The radical adducts that *β*-carotenoid forms with free radicals derived from linoleic acid are what give it its antioxidant properties. The highly unsaturated *β*-carotene model is attacked by the free radical linoleic acid. By scavenging the linoleate-free radical and other free radicals generated in the system, antioxidants can lessen the degree of *β*-carotene bleaching [23]. Accordingly, the samples without an antioxidant have a rapid decline in absorbance, whereas those with an antioxidant maintain their color for a longer period of time. Table 2 displays the antioxidant activity of freshly produced compounds as a percentage. Additionally, the 50 percent inhibitory concentration (IC_50_) for freshly synthesized compounds’ DPPH activity was calculated (Table 3). According to the two methods used and when compared to ascorbic acid, a common antioxidant, the synthesized compounds demonstrated promising radical scavenging activities. According to the findings, the majority of the compounds showed strong radical scavenging properties at lower concentrations. However, in every instance, a progressive rise in activity was seen as the concentration of the test substances increased.

Compounds **2–6** and **9** were thought to have moderate activity, whereas **7** and **8** showed good antioxidant characteristics which may be attributed to the additional (-N-Me^,^s) branching effects. It is obvious that the antioxidant properties were strengthened by the presence of phosphorus heterocycles. The studied compounds **11**, **12,** and **14** exhibited high potentials as antioxidant agents; in particular, oxathiaphosphininoquinolinone **14** had the highest values, as opposed to compound **13**, which had the lowest activity. This might be the result of efficient conjugation as well as the presence of the electron-donor group methoxy.

## 3. Experimental

### 3.1. Instruments and Reagents

An automated melting point technique called Optimelt was used to calculate the uncorrected melting points. The IR spectra were captured using a Perkin-Elmer 1800 Series FT-IR spectrometer. The samples were assessed using thin films on KBr plates. The ^1^H- and ^13^C-NMR spectra were captured at an ambient temperature in base-filtered DMSO-*d*_6_ as a solvent and TMS (δ) as an internal standard on a Bruker Avance III-400 MHz instrument running at 400 MHz for protons, 100 MHz for carbon, and 240 MHz for phosphorus nuclei. Molecular formulas were recorded with a Gas Chromatographic GCMSqp 1000 ex Shimadzu instrument at 70 eV. Analytical thin-layer chromatography (TLC) was carried out using aluminum-backed, 0.2-mm-thick silica gel 60 F254 plates. Using a UV lamp with a wavelength of 254 nm and/or by giving the plates a suitable dip and then heating them, eluted plates may be observed. These dips contained either potassium permanganate, potassium carbonate, 5 percent sodium hydroxide aqueous solution, phosphomolybdic acid, ceric sulfate, sulfuric acid (conc.), and water (37.5 g, 7.5 g, 37.5 g, 720 mL) (3 g, 20 g, 5 mL, 300 mL). The retardation factor (R*_f_*) values have been rounded to the nearest whole number. With silica gel 60 (40–63 μm) as the stationary phase, column chromatographic separations were carried out using AR- or HPLC-grade solvents. Starting materials, reagents, drying agents, and other inorganic salts could all be purchased commercially and used according to instructions.

### 3.2. Synthesis

#### 3.2.1. Synthesis of Diethyl 5,6-dihydro-6-methyl-4,5-dioxo-4*H*-pyrano[3,2-*c*]quinolin-2-yl-2-phosphonate (**2**)

A mixture of 3-[(*E*)-3-(Dimethylamino)-2-propenoyl]-4-hydroxy-1-methyl-2(1*H*)-quinolinone (**1**) (1.36 g, 5 mmol) and diethyl phosphite (0.7 mL, 5 mmol) in dry toluene (20 mL) containing a few drops of DBU as a catalyst, was heated under reflux for 6 h. The resulting light yellow solid was subjected to column chromatography to yield product **2**.

Recryst. Solvent: hexane; pale yellow crystals; R*_f_* = 0.31 (EtOAc/hexane, 1:3); yield 35%; mp 114–116 °C. IR (KBr), (*v* max, cm^−1^): 3050 (C−H_arom_), 2980 (C−H_aliph_), 1650 (C=O_pyrone_), 1615 (C=C), 1228 (P=O), 1026 (P−O−C). ^1^H-NMR (400 MHz, DMSO-*d*_6_): 1.20 (t, 6H, *J*=7.3 Hz, CH_3_), 3.56 (s, 3H, NCH_3quinolinone_), 4.15 (q, 4H, *J* = 7.3 Hz, OCH_2_), 6.20 (s, 1H, H−3), 7.20–7.85 (m, 4H, Ar–H_quinolinone_); ^13^C-NMR (100 MHz, DMSO-*d*_6_): 17.5 (CH_3_), 26.0 (NCH3), 62.0 (OCH_2_), 110.2(C–4a), 111.5 (C-7), 114.0 (C-3), 118.0 (C–10a), 120.5 (C–9), 122.2 (C– 10), 131.6 (C–8), 132.6 (C–6a), 155.2 (C–10b), 156.2 (C=O_quinolinone_). 158.0 (d, *J* = 152 Hz, C-2), 170.5 (C=O_pyran_); ^31^P-NMR (240 MHz, DMSO-*d_6_*)): 21.1 ppm; MS (*m/z*, I %): 363 (M^+^, 10%). Anal. Calc. (%) for C_17_H_18_NO_6_P (363.09): C, 56.20; H, 4.99; N, 3.86. Found: C, 56.18; H, 4.95; N, 3.82.

#### 3.2.2. General Procedure for the Synthesis of Compounds **3**, **4**, **5**, **6**, and **9**

The phosphonate reagent (3.0 mmol) namely ethyl (diethoxyphosphoryl) acetate, diethyl 2-amino-2-thioxoethylphosphonate, diethyl vinylphosphonate, diethyl cyanomethylphosphonate, or *O*,*O*-diethyl dithiophosphoric acid was added to a solution of enaminone **1** (0.7 g, 2.5 mmol) dissolved in a mixture of acetic acid (15 mL)/conc H_2_SO_4_ (0.7 mL) and the reaction mixture was further heated at 100 °C for the appropriate time (10–15 h, TLC). The product mixture was cooled and poured into ice water. The residue was subjected to column chromatography to afford the product. The solid product was collected, washed with dil ethanol (1/1 *v*/*v*), dried, and crystallized from the proper solvent to afford the corresponding phosphonate 3, 4, 5, 6, or 9.

#### 3.2.3. Diethyl [6-(4-hydroxy-1-methyl-2-oxo-1,2-dihydroquinolin-3-yl)-2-oxo-2*H*-pyran-3-yl]phosphonate (**3**)

Recryst. Solvent: ethanol; yellow crystals; R*_f_* = 0.30 (EtOAc/hexane, 1:2); yield 50%; mp 210–212 °C. IR (KBr), (*v*_max_, cm^−1^):3340 (OH), 3043 (C−H_arom_), 2900 (C−H_aliph_), 1670 (C=O), 1600, 1591 (C=C), 1250 (P=O), 1070, 1032 (P-O-C). ^1^H-NMR (400 MHz, DMSO-*d*_6_): 1.15 (t, 3H, *J* = 7.2 Hz, CH_3_), 1.30 (t, 3H, *J* = 7.2 Hz, CH_3_), 3.25 (s, 3H,NCH_3quinolinone_), 3.75−3.80 (m, 2H, OCH_2_), 4.13−4.15 (m, 2H, OCH_2_), 6.50 (d, 1H, *J* = 5.6 Hz, H−5′), 7.49–8.00 (m, 4H, Ar–H_quinolinone_ and H−4′), 10.70 (brs, 1H, OH). ^13^C-NMR (100 MHz, DMSO-*d*_6_):14.2 (CH_3_), 28.9 (N–CH_3_), 61.5 (OCH_2_), 98.0 (C−5′), 99.5 (C-3), 112.5 (C-8), 114.7 (C-4a), 116.0 (d, *J* = 174 Hz, C−3′), 117.5 (C-6), 120.0 (C-5), 121.5 (C-7), 129.2 (C-8a), 157.5 (C−4′), 160.5 (C-4), 162.6 (C=O_quinolinoe_), 165.7 (C−6′), 169.2 (C−2′). ^31^P-NMR (240 MHz, DMSO-*d_6_*): 24.2 ppm. MS (*m*/*z*, I %): 405 (M^+^, 20%). Anal. Calc. (%) for C_19_H_20_NO_7_P (405.34): C, 56.30; H, 4.97; N, 3.46; Found: C, 56.19; H, 4.92; N, 3.48.

#### 3.2.4. Diethyl[6-(4-hydroxy-1-methyl-2-oxo-1,2-dihydroquinolin-3-yl)-2-thioxo-1,2-dihydropyridin-3-yl]phosphonate (**4**)

Recryst. Solvent: DMF/EtOH; yellow solid; *R_f_* = 0.9 (acetone/hexane, 1:2); yield 55%; mp 187–188 °C. IR (KBr), (*v* max, cm^−1^): 3340 (OH), 3330 (NH), 3043 (C−H_arom_), 2900 (C−H_aliph_), 1670 (C=O), 1600, 1591 (C=C), 1230 (P=O), 1070, 1225 (C=S), 1025 (P-O-C), 1033 (C−O). ^1^H-NMR (400 MHz, DMSO-*d*_6_): 1.18 (t, 6H, *J* = 7.2 Hz, CH_3_), 3.50 (3H, s, NCH_3quinolinone_), 3.77 (m, 4H, OCH_2_), 6.93 (d, 1H, *J* = 5.6 Hz, H−5′), 7.30–8.20 (m, 4H, Ar–H and H−4′), 10.5 (brs, 1H, HN), 11.00 (brs, 1H, OH). ^13^C-NMR (100 MHz, DMSO-*d*_6_): 13.2 (CH_3_), 28.4 (N–CH_3_), 55.0 (OCH_2_), 99.2 (C−5′), 100.5 (C-3), 113.2 (C-8), 118.1 (C-4a), 119.5 (d, *J* = 174 Hz, C−3′), 123.0 (C-6), 125.5 (C-5), 128.4 (C-7), 131.4 (C-8a), 157.6 (C−4′), 164.0 (C-4), 166.4(C-2), 168.1 (C−6′), 174.7 (C=S). ^31^P-NMR (240 MHz, DMSO-*d_6_*): 26.4 ppm. MS (*m*/*z*, I %): 420 (M^+^, 30%). Anal. Calc. (%) for C_19_H_21_N_2_O_5_PS (420.09): C, 54.28; H, 5.03; N, 6.66; Found: C, 54.25; H, 5.00; N, 6.64.

#### 3.2.5. Diethyl[6-(4-hydroxy-1-methyl-2-oxo-1,2-dihydroquinolin-3-yl)-2*H*-pyran-3-yl]phosphonate (**5**)

Recryst. Solvent: DMF; Orange solid; R*_f_* = 0.4 (EtOAc/hexane, 1:2); yield 40%; mp 196-198 °C. IR (KBr), (*v* max, cm^−1^):3330 (OH), 3023 (C−H_arom_), 2950 (C−H_aliph_), 1660 (C=O), 1610, 1590 (C=C), 1253 (P=O), 1131 (P–O–C). ^1^H-NMR (400 MHz, DMSO-*d*_6_): 1.34 (t, 6H, *J* = 7.2 Hz, CH_3_), 3.38 (3H, s, NCH_3quinolinone_), 4.07 (m, 4H, OCH_2_), 5.00, 5.15 (2dd, *J*_HH_ = 7.6, 2H−2′), 6.55 (d, 1H, *J* = 5.6 Hz, H−5′), 7.25–8.22 (m, 4H, Ar–H_quinolinone_ and H−4′), 12.07 (brs, 1H, OH). ^13^C-NMR (100 MHz, DMSO-*d*_6_): 13.1 (CH_3_), 28.4 (N–CH_3_), 60.1 (OCH_2_), 61.8 (C−2′), 98.2 (C−5′), 99.8 (C-3), 118.1 (C-8), 119.7 (C-4a), 123.2 (d, *J* = 174 Hz, C−3′), 125.4 (C-6), 125.8 (C-5), 126.3 (C-7), 127.8 (C-8a), 157.9 (C−4′), 161.5 (C-4), 164.7 (C-2), 168.1 (C−6′). ^31^P-NMR (240 MHz, DMSO-*d_6_*): 25.5 ppm. MS (*m*/*z*, I %): 391 (M^+^, 40%). Anal. Calc. (%) for C_19_H_22_NO_6_P (391.12): C, 58.31; H, 5.67; N, 3.58; Found: C, 58.29; H, 5.66; N, 3.56.

#### 3.2.6. Diethyl [6-(4-hydroxy-1-methyl-2-oxo-1,2-dihydroquinolin-3-yl)-2-oxo-1,2-dihydropyridin-3-yl]phosphonate (**6**)

Recryst. Solvent: ethanol; Orange solid; R*_f_* = 0.6 (acetone/hexane, 1:1); yield 48%; mp 180–182 °C. IR (KBr), (*v* max, cm^−1^):3345 (OH), 3330 (NH), 3043 (C−H_arom_), 2980 (C−H_aliph_), 1720 (C=O_amide_), 1675 (C=O_quinolinone_), 1610, 1591 (C=C), 1220 (P=O), 1076 (P-O-C), 1030 (C−O). ^1^H-NMR (400 MHz, DMSO-*d*_6_): 1.13 (t, 6H, *J* = 7.0 Hz, CH_3_), 3.55 (3H, s, NCH_3quinolinone_), 3.83 (m, 4H, OCH_2_), 6.63 (d, 1H, *J* = 5.5 Hz, H−5′), 7.31–8.21 (m, 4H, Ar–H and H−4′), 10.21 (brs, 1H, HN), 11.56 (brs, 1H, OH). ^13^C-NMR (100 MHz, DMSO-*d*_6_): 15.5 (CH_3_), 26.4 (N–CH_3_), 64.0 (OCH_2_), 99.5 (C−5′), 105.8 (C-3), 114.0 (C-8), 117.1 (C-4a), 119.5 (d, *J* = 174 Hz, C−3′), 122.0 (C-6), 125.5 (C-5), 128.5(C-7), 131.5 (C-8a), 157.2 (C−4′), 164.5 (C=O _amide_) 166.0 (C-4), 168.1 (C-2), 172.0 (C−6′). ^31^P-NMR (240 MHz, DMSO-*d_6_*): 24.0 ppm. MS (*m*/*z*, I %): 420 (M^+^, 30%). Anal. Calc. (%) for C_19_H_21_N_2_O6P (404.11): C, 56.44; H, 5.23; N, 6.93; Found: C, 56.25; H, 5.20; N, 6.90.

#### 3.2.7. General Procedure for Synthesis of Compounds **7** and **8**

The phosphonate reagent (5.0 mmol), namely, diethyl cyanomethylphosphonate, and *O,O*-diethyl dithiophosphoric acid and enaminone 2 (1.4 g, 5 mmol) in absolute ethanol (20 mL) was maintained under reflux for 10 h. The solids formed at heat were filtered off and washed with water before being subjected to column chromatography to give the product 7 and 8, respectively.

#### 3.2.8. Diethyl-4-(dimethylamino)-1,2-dihydro-6-(1,2-dihydro-4-hydroxy-1-methyl-2-oxoquinolin-3-yl)-2-oxopyridin-3-yl-3-phosphonate (**7**)

Recryst. Solvent: ethanol; Orange solid; R*_f_* = 0.5 (acetone/hexane, 1:1); yield 62%; mp 195–198 °C. IR (KBr), (*v* max, cm^−1^):3340 (OH), 3310 (NH), 3040 (C−H_arom_), 2950 (C−H_aliph_), 1710 (C=O_amide_), 1665 (C=O_quinolinone_), 1600, 1590 (C=C), 1210 (P=O), 1070 (P-O-C). ^1^H-NMR (400 MHz, DMSO-*d*_6_): 1.30 (t, 6H, *J* = 7.6 Hz, CH_3_), 2.56 (s, 3H, NCH_3_), 2.84 (s, 3H, NCH_3_),3.58 (3H, s, N–CH_3_), 4.07 (m, 4H, OCH_2_), 7.25–8.20 (m, 4H, Ar–H and H−5′), 10.20 (brs, 1H, HN), 11.10 (brs, 1H, OH). ^13^C-NMR (100 MHz, DMSO-*d*_6_): 13.5 (CH_3_), 28.5 (NCH_3quinolinone_), 43.4 (NCH_3_), 46.1 (NCH_3_), 60.1 (OCH_2_), 95.75 (C−5′), 105.0 (C-3), 118.22 (C-8), 118.3 (C-4a), 119.3 (d, *J* = 170 Hz, C−3′), 123.4 (C-6), 125.5 (C-5), 125.8 (C-7), 130.3 (C-8a), 156.1 (C−4′), 161.5 (C=O _amide_) 163.3 (C-4), 165.3 (C-2), 166.0 (C−6′), ^31^P-NMR (240 MHz, DMSO-*d_6_*): 25.07 ppm. MS (*m*/*z*, I %): 447 (M^+^, 10%). Anal. Calc. (%) for C_21_H_26_N_3_O6P (447.16): C, 56.37; H, 5.86; N, 9.39; Found: C, 56.35; H, 5.84; N, 9.37.

#### 3.2.9. 3-[4-(dimethylamino)-2-ethoxy-2-oxido-4*H*-1,3,2-oxathiaphosphinin-6-yl]-4-hydroxy-1-methylquinolin-2(1*H*)-one (**8**)

Recryst. Solvent: ethanol; yellow solid; R*_f_* = 0.4 (acetone/hexane, 1:1); yield 40%; mp 125−127 °C. IR (KBr), (*v* max, cm^−1^): 3336 (OH), 3064 (C−H_arom_), 2943, 2911 (C−H_aliph_), 1598 (C=C), 1052 (C−O). ^1^H-NMR (400 MHz, DMSO-*d*_6_): 1.35 (t, 3H, *J* = 7.6 Hz, CH_3_), 3.00 (s, 3H, NCH_3_), 3.10 (s, 3H, NCH_3_), 3.43 (s, 3H, NCH_3quinolinone_), 4.04 (q, 2H, *J* = 7.6 Hz, OCH_2_), 6.10 (d, 1H, *J* = 7.2 Hz, H−4′), 6.25 (d, 1H, *J* = 7.2 Hz, H−5′), 6.80 (d, 1H, *J* = 7.6 Hz, H−5_quinolinone_), 7.18 (t, 1H, *J* = 7.2 Hz, H−7_quinolinone_), 7.29(m, 1H, H−6_quinolinone_), 8.05 (dd, 1H, *J* = 8.0 and 1.6 Hz, H−8_quinolinone_), 10.50 (brs, 1H, OH). ^13^C-NMR (100 MHz, DMSO-*d*_6_): 14.7 (*C*H_3_), 28.5 (q, N*C*H_3quinolinone_), 41.1 (N*C*H_3_), 44.1 (N*C*H_3_), 62.1 (O*C*H_2_), 65.3 (C−4′), 93.1 (C−5′), 104.3 (s, C-3), 114.3 (s, C-4a), 115.0 (d, C-8), 122.0 (d, C-6), 127.1 (d, C-5), 137.4 (d, C-7), 139.6 (s, C-8a), 144.1 (s, C-4),152.9 (C−6′), 161.0 (s, C-2). ^31^P-NMR (240 MHz, DMSO-*d_6_*): 35.5 ppm. MS (*m/z*, I %): 396 (M^+^, 14%). Anal. Calc. (%) for C_17_H_21_N_2_O_5_PS (396.09): C, 51.51; H, 5.34; N, 7.07; S, 8.09 %. Found: C, 51.50; H, 5.30; N, 7.00; S, 8.02.

#### 3.2.10. 3-(2-ethoxy-2-oxido-4*H*-1,3,2-oxathiaphosphinin-6-yl)-4-hydroxy-1-methylquinolin-2(1H)-one (**9**)

Recryst. Solvent: ethanol; Pale Yellow solid; R*_f_* = 0.3 (EtOAc/hexane, 1:2); yield 45% yield; mp 145−148 °C. IR (KBr), (*v* max, cm^−1^): 3332 (OH), 3060 (C−H_arom_), 2940, 2915 (C−H_aliph_), 1595 (C=C), 1050 (C−O). ^1^H-NMR (400 MHz, DMSO-*d*_6_): ^1^H-NMR (400 MHz, DMSO-*d_6_*): 1.30 (t, 3H, *J* = 7.6 Hz, CH_3_), 3.10 (3H, s, N–CH**_3_**_quinolinone_), 3.85 (q, 2H, *J* = 7.6 Hz, OCH_2_), 4.33, 4.35 (2dd, *J*_HH_ = 7.5, 2H, H_2_C(4) _oxathiaphosphinine_), 7.35 (t, *J*_HH_ = 7.2, 1H, HC(5)/_oxathiaphosphinine_), 7.46 (d, 1H, *J* = 7.6 Hz, H−5 _quinolinone_), 7.81 (t, 1H, *J* = 7.2 Hz, H−7 _quinolinone_), 7.89 (m, 1H, H−6 _quinolinone_), 8.07 (dd, 1H, *J* = 8.0 and 1.6 Hz, H−8 _quinolinone_), 10.60 (s, 1H, OH). ^13^C-NMR (100 MHz, DMSO-*d*_6_): 12.0 (CH_3_), 28.6 (q, N–CH_3_), 61.0 (OCH_2_), 64.0 (C−4′), 95.8 (C−5′), 106.2 (s, C-3), 111.8 (s, C-4a), 119.8 (d, C-8), 121.7 (d, C-6), 124.8 (d, C-5), 128.6 (d, C-7), 129.4 (s, C-8a), 139.8 (s, C-4), 155.8 (C−6′), 161.3 (s, C-2). ^31^P-NMR (240 MHz, DMSO-*d_6_*): 30.0 ppm. MS (*m/z*, I %): 353 (M^+^, 10%). Anal. Calc. (%) for C_15_H_16_NO_5_PS (353.05): C, 50.99; H, 4.56; N, 3.96; Found: C, 50.905; H, 4.52; N, 3.93.

#### 3.2.11. General Procedure for Synthesis of Compounds **11** and **12**

A magnetically agitated solution of the enaminone **1** (1.4 g, 5 mmol) in dry dioxane (20 mL) containing a catalytic quantity of triethylamine (1.4 mL, 10 mmol) was dropwise added to the solution of *P*-phenylphosphonic diamide (0.79 g, 5 mmol) in dry dioxane (5 mL) for 30 min at 5–10 °C. For four hours, the reaction mixture was held at 70 °C. The resulting orange solid was filtered out, washed with water, and then column chromatographed to produce product **11**. Under low pressure, the reaction filtrate was concentrated. The resultant oily residue was heated under reflux for 20 min after being diluted with distilled water (10 mL). Column chromatography was used to produce product **12** from the produced solid.

#### 3.2.12. 4-hydroxy-1-methyl-3-(2-oxido-2-phenyl-1,2-dihydro-1,3,2-diazaphosphinin-4-yl)quinolin-2(1*H*)-one (**11**)

Recryst. Solvent: ethanol; orange crystals; R*_f_* = 0.4 (acetone/hexane, 1:2); yield 80%; mp 125–127 °C. IR (KBr), (v max, cm-1): 3230–3175 (br, NH), 3069 (C–H_arom_), 1680 (C=O), 1650 (C=N), 1597, 1582 (C=C), 1250 (P=O). ^1^H-NMR (400 MHz, DMSO-*d_6_*): 3.34 (3H, s, N–CH**_3_**), 6.94 (dd, *J*_HH_ = 7.4, 1H, HC(5′)_diazaphosphinine_), 7.04 (dd, *J*_HH_ = 7.4, 1H, HC(6′)_diazaphosphinine_), 7.28–8.15 (m, 9H, Ar–H), 10.81 (brs, 1H, OH), 12.13 (brs, 1H, NH). ^13^C-NMR (100 MHz, DMSO-*d_6_*): 28.8 (N–CH_3_), 99.4 (C-3), 103.2 (C–4_diazaphosphinine_), 110.2 (C-8), 118.2 (C-4a), 119.3 (C-6), 123.4 (C-5), 125.8 (C-7), 130.6 (C–3,5_phenyl_), 131.6 (C–2,6_phenyl_, 135.3 (C-8a), 138.3 (C–4_phenyl_), 139.2 (C–1_phenyl_), 152.8 (C–5_diazaphosphinine_), 157.3 (C–4_diazaphosphinine_), 161.0 (C-4), 164.5 (C=O_quinolinone_). ^31^P-NMR (240 MHz, DMSO-*d_6_*): 31.50 ppm. MS (*m*/*z*, I %): 365 (M^+^, 20%). Anal. Calc. (%) for C_19_H_16_N_3_O_3_P (405.34): C, 62.47; H, 4.41; N, 11.50; Found: C, 62.43; H, 4.39; N, 11.55.

#### 3.2.13. 4-hydroxy-1-methyl-3-(2-oxido-2-phenyl-3,4-dihydro-2*H*-1,3,2-oxazaphosphinin-6-yl)quinolin-2(1*H*)-one (**12**)

Recryst. Solvent: ethanol; Pale Yellow solid; R*_f_* = 0.3 (EtOAc/hexane, 1:1); yield 75%; mp 115–118 °C. IR (KBr), (v max, cm^−1^): 3200-3150 (br, NH), 3069 (C–H_arom_), 1680 (C=O), 1650 (C=N), 1597, 1582 (C=C), 1253 (P=O), 1131 (P-O-C). ^1^H-NMR (400 MHz, DMSO-*d_6_*): 3.30 (3H, s, N–CH**_3_**_quinolinone_), 4.05, 4.17 (2dd, *J*_HH_ = 7.5, 2H, H_2_C(4)/_oxazaphosphinine_), 6.55 (t, *J*_HH_ = 7.2, 1H, HC(5)/_oxazaphosphinine_), 6.95–8.18 (m, 9H, Ar–H), 9.80 (brs, 1H, OH), 12.50 (brs, 1H, NH). ^13^C-NMR (100 MHz, DMSO-*d_6_*): 28.6 (N–CH_3_), 68.6 (C−4/), 94.4 (C−5/), 100.5 (C-3), 112.2 (C-8), 118.6 (C-4a), 122.1 (C-6), 125.8 (C-5), 126.6 (C-7), 126.7 (C–3,5_phenyl_), 127.0 (C–2,6_phenyl_), 127.2 (C-8a), 129.3 (C–4_phenyl_), 131.3 (d, *J* = 117 Hz, C–1_phenyl_), 151.0 (C−6/), 154.7 (C-4), 162.5(C=O_quinolinone_). ^31^P-NMR (240 MHz, DMSO-*d_6_*): 28.50 ppm. MS (*m*/*z*, I %): 368 (M^+^, 20%). Anal. Calc. (%) for C_19_H_17_N_2_O_4_P (368.32): C, 61.96; H, 4.65; N, 7.61; Found: C, 61.99; H, 4.70; N, 7.58.

#### 3.2.14. Synthesis of 6-methyl-4-thioxo-4*H*-thiopyrano[3,2-c]quinolin-5(6*H*)-one (**13**)

A mixture of phosphorus pentasulfide (1.11 g, 5 mmol) and compound 2 (1.4 g, 5 mmol) in dry toluene (40 mL), was heated under reflux for 7 h. The reaction mixture was concentrated into its half volume and left to cool. The formed solid was filtered off and recrystallized from ethanol to give an orange crystalline solid at 70% yield; mp 260–262 °C. IR (KBr), (*v* max, cm^−1^): 3060 (C−H_arom_), 1615, 1600 (C=C), 1240 (C=S), 1640 (C=O_quinolone_). ^1^H-NMR (400 MHz, DMSO-*d_6_*): δ = 3.28 (3H, s, N–CH_3_), 5.56 (d, 1 H, ^3^*J*_2,3_ = 6.0 Hz, H-3), 7.14 (1H, ddd, ^3^*J* = 8.0, 7.8 Hz, ^4^*J* = 1.0 Hz, H-9), 7.32 (1H, dd, ^3^*J* = 8.4 Hz, ^4^*J* = 1.0 Hz, H-7), 7.46 (1H, ddd, ^3^*J* = 8.6, 7.8 Hz, ^4^*J* = 1.4 Hz, H-8), 7.68 (d, 1 H, H-2), 8.05 (1H, dd, ^3^*J* = 8.0 Hz, ^4^*J* = 1.4 Hz, H-10); ^13^C-NMR (100 MHz, DMSO-*d*_6_): δ 28.6 (N–CH_3_), 114.0 (4a), 116.6 (C-10a), 117.4 (C-7), 122.5 (C-9), 127.1 (C-10), 127.6 (C-8), 131.2 (C-3), 135.0 (s, C-6a), 136.1 (C-10b), 155.0 (C-2), 158.6 (C-5), 175.10 (C=S). MS (*m*/*z*, I %): 259 (M^+^, 30%). Anal. Calc. (%) for C_13_H_9_NOS_2_ (259.01): C, 60.20; H, 3.50; N, 5.40; S, 24.73 Found: C, 60.18; H, 3.52; S, 24.71.

#### 3.2.15. Synthesis of 3-(4-(dimethylamino)-2-(4-methoxyphenyl)-2-sulfanylene-4*H*-1,3,2-oxathiaphosphinin-6-yl)-4-hydroxy-1-methylquinolin-2(1*H*)-one (**14**)

Lawesson’s reagent (1 g, 5 mmol) was added to a solution of compound **1** (1.4 g, 5 mmol) in dry toluene (20 mL). The mixture was heated under reflux for 6 h. The formed solid was filtered off and crystallized from diluted ethanol to give a pale brown solid at 60% yield; mp 105−108 °C. IR (KBr), (*v* max, cm^−1^): 3350 (OH), 3030 (C−H_arom_), 2970, 2820 (C−H_aliph_), 1600, 1565 (C=C), 1020 (C−O). ^1^H-NMR (400 MHz, DMSO-*d*_6_): 2.90 (s, 3H, NCH_3_), 3.15 (s, 3H, NCH_3_), 3.32 (s, 3H, NCH_3_), 3.80 (s, 3H, OCH_3_), 5.50 (d, 1H, *J* = 7.2 Hz, H−4′), 6.30 (d, 1H, *J* = 7.6 Hz, H−5′), 7.05 (m, 2H, H−3″,5″_aryl_), 7.12 (1H, ddd, ^3^*J* = 8.0, 7.8 Hz, ^4^*J* = 1.0 Hz, H-6), 7.35 (1H, dd, ^3^*J* = 8.4 Hz, ^4^*J* = 1.0 Hz, H-8), 7.41 (1H, ddd, ^3^*J* = 8.6, 7.8 Hz, ^4^*J* = 1.4 Hz, H-7),7.60 (d, 2H, *J* = 8.0 Hz, H−2″,6″_aryl_), 7.85 (1H, dd, ^3^*J* = 8.0 Hz, ^4^*J* = 1.4 Hz, H-5),11.00 (brs, 1H, OH). ^13^C-NMR (100 MHz, DMSO-*d*_6_): 29.5 (q, N–CH_3_), 32.13 (NCH_3_), 33.85 (NCH_3_), 52.3 (OCH_3_), 64.16 (C−4′), 101.52 (C−5′), 105.13 (s, 3), 111.1 (s, C-4a), 114.0 (d, C-8), 114.2 (C−3″,5″_aryl_), 122.5 (d, *J* = 168 Hz, C−1″_aryl_), 123.3 (d, C-6), 127.6 (d, C-5), 131.7 (d, C-7),137.9 (C−2″,6″_aryl_), 138.5 (s, C-8a), 141.3 (s, C-4),151.1 (C−6′), 155.6 (C−4″_aryl_),162.8 (s, C-2). ^31^P-NMR (240 MHz, DMSO-*d_6_*): 28.5 ppm. MS (*m/z*, I %): 393 (M^+^, 27%). Anal. Calc. (%) for C_22_H_23_N_2_O_4_PS_2_ (474.08): C, 55.68; H, 4.89; N, 5.90; Found: C, 55.59; H, 4.85; N, 5.92.

### 3.3. Antioxidant Activity

Activity to scavenge DPPH radicals. According to the described method, the DPPH (2,2-diphenyl-1-picrylhydrazyl) radical scavenging effect was assessed [22,23]. Four milliliters of a 0.004 percent (*w*/*v*) ethanol solution of DPPH were mixed with one milliliter of the solutions of the test compounds at different concentrations (50, 75, and 100 g/mL) in ethanol. The tubes were then incubated for 30 min at RT in the pitch-black chamber. A compound-free DPPH blank was created, and ethanol was employed for the baseline correction. Using a UV-Vis spectrophotometer, changes (decreases) in the absorbance at 517 nm were measured. Formula: percent radical scavenging activity = (AB-AA)/AB 100, where AB = absorption of the blank and AA = absorption of the tested drug. The radical scavenging activities were represented as the inhibition percentage. Ascorbic acid’s radical scavenging capacity was also evaluated in comparison to that of other synthetic compounds. It was determined what chemical concentration would provide 50% inhibition (IC_50_). Assay for antioxidant activity using -carotene and linoleic acid. Each substance was individually added to the -carotene-linoleic acid model system at final concentrations of 50, 75, and 100 g/mL, and the activity was measured spectrophotometrically at 470 nm. The substrate suspension was formed by adding linoleic acid and -carotene to a lidded, round-bottomed flask containing Tween-40 (600 mg) and -carotene (4 mg dissolved in 5 mL of chloroform) (60 mL). The rotary evaporator at 40 °C was used to thoroughly evaporate the chloroform under a vacuum. The resultant mixture was well-mixed and diluted with oxygenated water (30 mL), and triple-distilled water was added to the emulsion (120 mL). The aliquot (4 mL) was then transferred to other test tubes with stoppers that contained different chemicals in distilled ethanol. Emulsion and distilled ethanol (1 mL) were used as the control (4 mL). For comparison, an internal standard ascorbic acid solution at the same concentration was also examined. With distilled water, there was no correction needed. Each test tube’s absorbance was first measured at 470 nm at zero time (t = 0) after the emulsion was applied, and then it was measured every 30 min up to three hours (t = 180) after that. Between readings, the tubes were submerged in a water bath set to 50 °C. The following formula was used to calculate the percentage of the antioxidant activity of each component in relation to the photooxidation of -carotene:% antioxidant activity = 100 × {1 − (Ao − At/Aoo − Ato)} (1)
where:Ao = the sample’s initial absorbance. (t = 0 min)At is the sample’s absorbance at time t. (t = 180 min)Aoo = the control’s initial absorbance. (t = 0 min)Ato is the absorption of control at time t. (t = 180 min)

## 4. Conclusions

Starting with 3-[(*E*)-3-(Dimethylamino)-2-propenoyl]-4-hydroxy-1-methy (**1**) and certain phosphorus reagents in various solvents, simple procedures were developed to synthesize new organophosphorus compounds. We anticipate that other researchers who are looking for novel synthetic fragments with distinctive properties for medicinal chemistry will find our method valuable. Significant antioxidant capabilities were demonstrated by the synthesized phosphorus heterocyclic compounds, which should spur further study in a number of areas related to pharmaceutical research and medical treatments.

## Data Availability

All data are available in the manuscript and from the corresponding author (M.S.) upon request.

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
