# Peer review of "Synthesis, Structural Determination, and Antioxidant Activities of Acyclic and Substituted Heterocyclic Phosphonates Linearly Linked 4-hydroxy-2(1H)-quinolinone"

_molecules, 2022, doi:10.3390/molecules27185960_

Round 1
Reviewer 1 Report
The manuscript by Hassan et al. reports synthesis, structural determination and antioxidant activities of acyclic and substituted heterocyclic phosphonates linearly linked 4-hydroxy-2(1H)-quinolinone. They synthesized new organo phosphorous compounds by reaction of 3-[(E)-3-(Dimethylamino)-2-propenoyl]-4-hydroxy-1-methyl-2(1H)-quinolinone (1) with certain phosphorus reagents in various solvents and compound 7 and 8 showed good antioxidant activity.
I recommend publication of this manuscript in Molecules subject to following modifications.
1) I would suggest to make abstract concise and also include antioxidant activity.
2) On page 2 in Results and discussion write complete name of compound 1 and in line 5 from bottom it should be compound 2 not 3.
3) On page 4, line 8 from top, yield of compound 5 should be 40% not 55%. Line 2 from bottom it should be compound 1 not 2.
4) In scheme 3, the starting material number is 1 not 2 and same on page 6, line 4 from bottom.
5) On page 7 and 8, scheme 4 and 5, correct numbering of compound 1 and 2.
6) On page 12, in experimental section, correct the Molecular formula and mass.
7) On page 16, in conclusion replace organ with organo or some other word.
Author Response
Synthesis, structural determination and antioxidant activities of acyclic and substituted heterocyclic phosphonates linearly linked 4-hydroxy-2(1H)-quinolinone.
Mohamed M. Hassan a* and Mona H. Alhalafi b
a Chemistry Department, Faculty of Education, Ain Shams University, Roxy, 11711 Cairo, Egypt
b Department of Chemistry, College of Science Al-zulfi, Majmaah University, P.O. 66, Al-Majmaah 11952, Saudi Arabia
* Correspondence: author: Mohamed M. Hassan - E-mail: mmhassan121@yahoo.com, The Orcid ID (https://orcid.org/0000- 0002-7496-4174), Mona H. Alhalafi - E-mail: m.alhalafi@mu.edu.s
Dear Editor-In-Chief,
I would like to thank you and the reviewers about evaluation of my manuscript and the valuable comments that really improve the manuscript and all my future manuscripts.
Each change made outlined (point by point) as raised in the reviewer comments with providing a suitable rebuttal to each reviewer comment not addressed. The corrections will appear in yellow background in the revised manuscript.
Reviewer#1Comments and Response
I recommend publication of this manuscript in Molecules subject to following modifications.
Reviewer Comment 1.
Would suggest to make abstract concise and also include antioxidant activity.
â–ºResponse: As our research focuses on synthesis and structural determination of acyclic and substituted heterocyclic phosphonates, which gave priority to detailing that in the abstract with concise as possible as we can with short reference to the antioxidant activities
Reviewer Comment 2.
On page 2 in Results and discussion write complete name of compound 1 and in line 5 from bottom it should be compound 2 not 3.
â–ºResponse: Edited and supplied the necessary corrections
Reviewer Comment 3.
On page 4, line 8 from top, yield of compound 5 should be 40% not 55%. Line 2 from bottom it should be compound 1 not 2.
â–ºResponse: Edited and supplied the necessary corrections
Reviewer Comment 4.
In scheme 3, the starting material number is 1 not 2 and same on page 6, line 4 from bottom.
â–ºResponse: Edited and supplied the necessary corrections
Reviewer Comment 5.
On page 7 and 8, scheme 4 and 5, correct numbering of compound 1 and 2.
â–ºResponse: Edited and supplied the necessary corrections
Reviewer Comment 6.
On page 12, in experimental section, correct the Molecular formula and mass.
â–ºResponse: Edited and supplied the necessary corrections
Reviewer Comment 7.
On page 16, in conclusion replace organ with organo or some other word.
â–ºResponse: Edited and supplied the necessary corrections

Reviewer 2 Report
This manuscript described the study on the chemical reactions of 3-[(E)-3-(dimethylamino)-2-propenoyl]-4-hydroxy-1-methy-2(1H)-quinolinone and different organo phosphours with the subsequent identification of the products based on the IR, MS and NMR data. Moreover, the reaction mechanism for these reactions were provided, which seemed reasonable. However, revisions are necessary before this manuscript was accepted.
1. Please delete the dot in the Title.
2. Please correct the name of the compound 1 as ‘3-[(E)-3-(dimethylamino)-2-propenoyl]-4-hydroxy-1-methy-2(1H)-quinolinone’ throughout the whole manuscript.
3. ‘Diethyl vinylphosphonate’ → ‘diethyl vinylphosphonate’
4. Please add the reference ‘Eur. J. Med. Chem. 2019, 165, 59-79.’ in the first paragraph.
5. The Ref. [10] was likely wrongly cited in the sentence ‘ In our most recent work, we created bioactive phosphorus compounds with heterocycles.10,11’
6. ‘Product 2's…’ → ‘Product 2's…’
7. ‘d 21.6 ppm’ → ‘δ 21.6 ppm’. Please check and correct this kind typo errors throughout the whole manuscript, such as ‘d 118.4ppm’.
8. For structure elucidation, the high resolution MS data are necessary. Please add them in the manuscript.
9. In the Synthesis, the eluting solvents for chromatography should be added.
10. Please pay attention to the font styles, such as ‘N–CH3’, ‘M+’, ‘1 H-NMR’, ‘13C-NMR’, ‘DMSO-d6’, ‘C22H23N2O4PS2’, as well as typo mistakes, such as ‘0C’, ‘ (2dd, JHH = 7.5, 2H...’, ‘23,24 Four millilitres…’, ‘IC50’.
11. Please provide the experimental procedure for compound 9.
12. As shown in the figures, compound 8 was a chiral compound. As reported, this compound was obtained as crystals. Did you perform the X-ray diffraction experiment? This experiment could help detect the purity as well as establish the configuration.
13. Please revise the references format according to ‘Instructions for Authors’. By the way, please update the literature information, such as Ref. [13].
14. As the experimental procedure and spectroscopic data of the synthesized compounds and the results of bioassay are recorded in the manuscript, only NMR, UV, IR, and MS spectra are required in the Supplementary Information.
Author Response
Synthesis, structural determination and antioxidant activities of acyclic and substituted heterocyclic phosphonates linearly linked 4-hydroxy-2(1H)-quinolinone.
Mohamed M. Hassan a* and Mona H. Alhalafi b
a Chemistry Department, Faculty of Education, Ain Shams University, Roxy, 11711 Cairo, Egypt
b Department of Chemistry, College of Science Al-zulfi, Majmaah University, P.O. 66, Al-Majmaah 11952, Saudi Arabia
* Correspondence: author: Mohamed M. Hassan - E-mail: mmhassan121@yahoo.com, The Orcid ID (https://orcid.org/0000- 0002-7496-4174), Mona H. Alhalafi - E-mail: m.alhalafi@mu.edu.s
Dear Editor-In-Chief,
I would like to thank you and the reviewers about evaluation of my manuscript and the valuable comments that really improve the manuscript and all my future manuscripts.
Each change made outlined (point by point) as raised in the reviewer comments with providing a suitable rebuttal to each reviewer comment not addressed. The corrections will appear in yellow background in the revised manuscript.
Reviewer#2Comments and Response
Reviewer Comment 1.
Please delete the dot in the Title.
â–ºResponse: Dot in the Title was deleted
Reviewer Comment 2.
Please correct the name of the compound 1 as ‘3-[(E)-3-(dimethylamino)-2-propenoyl]-4-hydroxy-1-methy-2(1H)-quinolinone’ throughout the whole manuscript.
â–ºResponse: the name of the compound 1 was corrected as ‘3-[(E)-3-(dimethylamino)-2-propenoyl]-4-hydroxy-1-methy-2(1H)-quinolinone’ throughout the whole manuscript.
Reviewer Comment 3.
‘Diethyl vinylphosphonate’ → ‘diethyl vinylphosphonate’
â–ºResponse: ‘Diethyl vinylphosphonate’ was corrected in the abstract to be ‘diethyl vinylphosphonate’
Reviewer Comment 4.
The Ref. [10] was likely wrongly cited in the sentence ‘ In our most recent work, we created bioactive phosphorus compounds with heterocycles.10,11’
â–ºResponse: The sentence has been modified to become ‘In recent studies conducted by us and others, bioactive phosphorus compounds containing heterocycles were produced’, so the Ref. [10] and [11] were well cited
Reviewer Comment 5.
‘Product 2's…’ → ‘Product 2's…’
â–ºResponse: ‘Product 2's was corrected to be Product 2's…’
Reviewer Comment 6.
‘d 21.6 ppm’ → ‘δ 21.6 ppm’. Please check and correct this kind typo errors throughout the whole manuscript, such as ‘d 118.4ppm’.
â–ºResponse: This kind typo errors was checked throughout the whole manuscript.
Reviewer Comment 7.
In the Synthesis, the eluting solvents for chromatography should be added.
â–ºResponse: The eluting solvents for chromatography were added
Reviewer Comment 8.
Please pay attention to the font styles, such as ‘N–CH3’, ‘M+’, ‘1 H-NMR’, ‘13C-NMR’, ‘DMSO-d6’, ‘C22H23N2O4PS2’, as well as typo mistakes, such as ‘0C’, ‘ (2dd, JHH = 7.5, 2H...’, ‘23,24 Four millilitres…’, ‘IC50’.
â–ºResponse: All the font styles were checked throughout the whole manuscript
Reviewer Comment 9.
Please provide the experimental procedure for compound 9.
â–ºResponse: the experimental procedure for compound 9 was. Provided
Reviewer Comment 10.
As shown in the figures, compound 8 was a chiral compound. As reported, this compound was obtained as crystals. Did you perform the X-ray diffraction experiment? This experiment could help detect the purity as well as establish the configuration.
â–ºResponse: Thank you very much for your constructive comment. We will take it into account in our future research
Reviewer Comment 11.
Please revise the references format according to ‘Instructions for Authors’. By the way, please update the literature information, such as Ref. [13].
â–ºResponse: All references format was revised and thee literature information updated such as [13].
